# Geochemical Characteristics and Toxic Elements in Alumina Refining Wastes and Leachates from Management Facilities

**DOI:** 10.3390/ijerph16071297

**Published:** 2019-04-11

**Authors:** Chunwei Sun, Jiannan Chen, Kuo Tian, Daoping Peng, Xin Liao, Xiyong Wu

**Affiliations:** 1Faculty of Geosciences and Environmental Engineering, Southwest Jiaotong University, Chengdu, Sichuan Province 611756, China; sunchunwei0310@gmail.com (C.S.); pdp0330@swjtu.cn (D.P.); wuxiyong@swjtu.edu.cn (X.W.); 2State-Province Joint Engineering Research Lab in Spatial Information Technology for High Speed Railway Operation Safety, Southwest Jiaotong University, Chengdu, Sichuan Province 611756, China; 3Department of Civil, Environmental, and Infrastructure Engineering, George Mason University, Fairfax, VA 22030, USA; ktian@gmu.edu

**Keywords:** red mud, alumina refining, leachate, geochemical characteristics, toxic elements, pH

## Abstract

A nationwide investigation was carried out to evaluate the geochemical characteristics and environmental impacts of red mud and leachates from the major alumina plants in China. The chemical and mineralogical compositions of red mud were investigated, and major, minor, and trace elements in the leachates were analyzed. The mineral and chemical compositions of red mud vary over refining processes (i.e., Bayer, sintering, and combined methods) and parental bauxites. The main minerals in the red mud are quartz, calcite, dolomite, hematite, hibschite, sodalite, anhydrite, cancrinite, and gibbsite. The major chemical compositions of red mud are Al, Fe, Si, Ca, Ti, and hydroxides. The associated red mud leachate is hyperalkaline (pH > 12), which can be toxic to aquatic life. The concentrations of Al, Cl^−^, F^−^, Na, NO_3_^2−^, and SO_4_^2−^ in the leachate exceed the recommended groundwater quality standard of China by up to 6637 times. These ions are likely to increase the salinization of the soil and groundwater. The minor elements in red mud leachate include As, B, Ba, Cr, Cu, Fe, Ni, Mn, Mo, Ti, V, and Zn, and the trace elements in red mud leachate include Ag, Be, Cd, Co, Hg, Li, Pb, Sb, Se, Sr, and Tl. Some of these elements have the concentration up to 272 times higher than those of the groundwater quality standard and are toxic to the environment and human health. Therefore, scientific guidance is needed for red mud management, especially for the design of the containment system of the facilities.

## 1. Introduction

Red mud is an insoluble residue produced by aluminum oxide (alumina) refining process with the characteristics of very fine particle size and high alkalinity [1]. In 2015, 274 million tons of bauxite ore were mined globally for alumina production, with Australia (33%), China (20%) and Brazil (16%) being the leading bauxite producers in the world [2]. Approximately 60 million tons of red mud is produced each year globally, with 6 million tons of red mud occurring in China. However, approximately 85% of the red mud is stored in the on-site reservoirs near the alumina refining plant. The reservoir is constructed with dams for containment purposes, and the red mud is treated by a natural settlement process [3].

Currently, manufacturers often adopt the chemical ore dressing process for alumina refining. This process uses the alkali method (sodium hydroxide or sodium carbonates) to produce alumina from bauxite and can be divided into the Bayer process, the sintering method, and the combined method [4,5]. The main components of the red mud are hematite (Fe_2_O_3_), calcite (CaCO_3_), cancrinite (Na_6_CaAl_6_Si_6_(CO_3_)O_24_·2H_2_O), hydrogarnet (Ca_3_AlFe(SiO_4_)(OH)_8_), tricalcium aluminate (Ca_3_Al_2_(OH)_12_), and sodalite ((Na_6_Al_6_Si_6_O_24_)·(2NaX or Na_2_X)) [4,6]. Red mud has strong alkalinity and salinity, with heavy metal elements, including arsenic (As), lead (Pb), zinc (Zn), copper (Cu), nickel (Ni), chromium (Cr), and vanadium (V). The salinity and alkalinity of red mud can affect plant growth and lead to deterioration of soil quality. The leakage or spill of red mud releases oxyanionic trace elements such as Cr, molybdenum (Mo), and V. These contaminants have been reported to be highly soluble under high pH conditions (material pH of red mud) in leaching tests [6,7], suggesting that red mud could be a potential sink of contaminants [8].

Red mud typically occupies land spaces for storage and further generates potential threats to the surrounding environment. Unlined or inappropriately lined red mud reservoirs could induce the leakage of leachate (i.e., bauxite liquor), leading soil swampiness and groundwater pollution [9,10]. Additionally, occasional dam failures damage infrastructure and contaminate the environment. For example, the catastrophic dam failure of the Ajkai Timfoldgyar Zrt red mud reservoir that occurred in Hungary released of 0.7 to 1 million m^3^ of caustic red mud into the Torna River, which resulted in the loss of lives and seriously contaminated soil and water [6,9,10,11].

Overall, the environmental risks are closely related to the soda content, alkalinity, and heavy metal content in the red mud. However, there is still a lack of knowledge regarding the mineralogy and chemical composition within red mud and its leachates in China. The objective of this study is to understand the potential contamination by red mud and its associated leachates from red mud management facilities. The chemical compositions of red mud from different resources and production process are summarized. As red mud leachate has essential effects on the environment, the leaching ability of red mud and its environmental impacts were studied through the analysis of the concentration of elements in leachates. Fresh red mud and its leachates were sampled from the major alumina manufacturers in China. The main chemical parameters, including pH, electrical conductivity (EC), redox potential (ORP), and elemental concentration, were evaluated through laboratory analytical methods.

## 2. Materials and Methods

### 2.1. Resources and Distribution of Bauxite

Bauxite is the primary aluminum ore composed of one or more aluminum hydroxides minerals, including gibbsite (Al(OH)_3_), boehmite (γ-AlO(OH)), diaspore (α-AlO(OH)), and impurities such as quartz (SiO_2_), hematite (α-Fe_2_O_3_), and rutile (TiO_2_) [4,12,13]. Bauxite resources in China are primarily distributed among seven provinces: Shanxi, Henan, Guangxi, Guizhou, Yunnan, Chongqing, and Shandong. Diaspore is the primary type of bauxite in China and contains 37.4~74.0% of Al_2_O_3_ and 3.5~32.2% of SiO_2_ [4,14,15,16,17,18]. The average Al_2_O_3_ content of bauxites in Guangxi varies between 52.3% and 62.4%, and the average Fe_2_O_3_ and SiO_2_ contents range between 15.0~24.5% and 3.5~8.3% (Table 1). However, the SiO_2_ contents (7.5~32.2%) in bauxite from Henan, Shandong, Shanxi, and Guizhou are higher than those from Guangxi, which result in a lower Al/Si ratio (1.2~9.4) than that of Guangxi bauxite (6.3~17.8).

The Al/Si ratio determines the alumina processing procedure. In general, Guangxi aluminum mines are bauxite deposits with a high Al/Si ratio (6.3-17.8). This type of bauxite can be utilized to extract alumina using the Bayer process. Shanxi, Shandong, Henan, Guizhou, and Sichuan aluminum mines, which account for over 98% of the reserves in China, are mainly bauxite deposits with a relatively low Al/Si ratio (3.3–9.4). These types of ores are middle/low-grade diaspore bauxites. They are usually challenging and energy consuming to process for alumina. Therefore, alternative methods modified from the traditional Bayer process, i.e., the sintering process and the Bayer-sintering combined process, have been developed for alumina refining [4].

### 2.2. The Sampling of Red Mud and Leachate

In this study, samples of red mud and leachates from five management facilities were collected from three provinces (i.e., Guangxi, Shangdong, and Henan) in China (Figure 1). The details of the sampling are summarized in Table 2. Fresh red mud samples GX-A1 and GX-A2, and leachate samples GX-A2-L and GX-A2-L were collected from the same manufacturer but two adjacent reservoirs (A1 and A2, respectively) in Pingguo County, Guangxi Province. The average annual temperature of Pingguo County is 21.5 °C, and the annual precipitation reaches 1500 mm. Fresh red mud sample GX-B and leachate sample GX-B-L were collected in Jingxi County, Guangxi Province. The average annual temperature of Jingxi County is 19.1 °C, and the annual precipitation reaches 1636 mm. Red mud sample SD-A and its leachate sample SD-A-L were collected in Zibo, Shandong. The average annual temperature of Zibo is 13.5 °C, and the annual precipitation reaches 650 mm. Red mud sample HN-A and its leachate sample HN-A-L were collected in Xingyang, Henan Province. The average annual temperature is 14.3 °C, and the annual precipitation reaches 645 mm. Fresh red mud was sampled after pressure filtration, and the leachate was collected from the drainage pipe of the red mud reservoir.

### 2.3. Bulk Chemical Analysis

Specimens were analyzed using X-ray fluorescence (XRF) quantitative analysis (Shimadzu XRF-1800, Kyoto, Japan). The analytical method for silicate rocks was used, and specimens were analyzed in duplicates. Five grams of each sample was air-dried for 24 h and ground to <0.075 mm with agate mortar and pestle. Loss on ignition tests were performed before XRF analysis by sintering samples at 900 °C using a muffle furnace (YSD-5-12T, Yaoshi Instrument Equipment Ltd., Shanghai, China). The LOI is used as an indicator for organic contents in the samples. XRF tests were conducted by fusing 0.9 g of calcined powder (i.e., specimen after LOI) with 1 g of NH_4_NO_3_ oxidizer and 9.0 g of lithium borate flux (50%/50% Li_2_B_4_O_7_-LiBO_2_) at 1050 °C into a flat molten glass disk. The specimens were then analyzed by XRF spectrometry.

### 2.4. Mineralogical Analysis

Quantitative X-ray diffraction analysis (Rigaku D/MAX-2005 X-ray diffractometer, Tokyo, Japan) was performed on the red mud samples to determine the major mineral phases. Specimens were placed in a desiccator for 24 h and ground to 0.075 mm with agate mortar and pestle. Cu Kα radiation was used, and each sample was placed in a 2-mm deep sample holder and scanned at 0.02° intervals between 5° to 50° 2θ with a 2 s dwell time. Samples were scanned within 48 h after desiccation to minimize crystal dehydration.

### 2.5. Hydrochemical Analysis

pH, EC, and ORP of leachate samples were recorded immediately after sampling. Leachate samples were then filtered through a 0.45-μm filter paper and preserved with trace-grade nitric acid (HNO_3_). The elemental concentrations of silver (Ag), aluminum (Al), As, beryllium (Be), boron (B), barium (Ba), calcium (Ca), cadmium (Cd), cobalt (Co), Cr, Cu, iron (Fe), mercury (Hg), potassium (K), lithium (Li), magnesium (Mg), manganese (Mn), Mo, sodium (Na), Ni, Pb, antimony (Sb), selenium (Se), silicon (Si), strontium (Sr), titanium (Ti), thallium (Tl), V, and Zn were determined by Inductively Coupled Plasma Mass Spectrometry (ICP-MS, Agilent Technologies 700 Series, Santa Clara, CA, USA) at Chengdu University of Science and Technology. Anions, including chloride (Cl^−^), fluoride (F^−^), nitrate (NO_3_^−^), and sulfate (SO_4_^2−^), were determined by ion chromatography (IC, Shimadzu HIC-SP, Kyoto, Japan) at Southwest Jiaotong University.

## 3. Results

### 3.1. Chemical and Mineral Compositions of Red Mud

The chemical and mineralogical compositions of red mud in China from this study and the literature are summarized in Table 3 and Table 4, respectively. The major chemical components of red mud are SiO_2_, TiO_2_, Al_2_O_3_, Fe_2_O_3_, CaO, and Na_2_O. These components comprise 69.5–98.3% of the total mass of the red mud. Na mainly comes from the sodium hydroxide (NaOH), which is used to treat bauxite, while Al, Si, Fe, Ti, and other elements are derived from the parental bauxite. The secondary components are K_2_O, MgO, P_2_O_5_, MnO, Cr_2_O_3_, and SO_3_. The main mineral phases include quartz (SiO_2_), calcite (CaCO_3_), dolomite (CaMg(CO_3_)_2_), hematite (Fe_2_O_3_), hydrogarnet (Ca_3_Al_2_(SiO_4_)_2_(OH)_4_), sodalite (Na_8_(Al_6_Si_6_O_24_)Cl_2_), anhydrite (CaSO_4_), cancrinite (Na_6_Ca_2_((CO_3_)_2_Al_6_Si_6_O_24_)·2H_2_O), and gibbsite (Al(OH)_3_).

The SiO_2_ content of red mud ranges from 8.4% to 32.5%. SiO_2_ is leached from kaolinite (Al_2_Si_2_O_3_(OH)_4_) in bauxite during alumina refining and exists in the form of hydrophane (SiO_2_·nH_2_O) and sodium silicate (Na_2_SiO_3_) [24]. In general, the red mud samples derived from Guangxi and Yunnan have lower SiO_2_ contents (8.4–11.9%) than those from Henan, Shandong, Shanxi and Guizhou (up to 32.5%) (Figure 2 and Table 3). This difference is mainly due to the different Si-Al ratio of bauxite ores. The Fe_2_O_3_ content of red mud ranges from 2.5% to 37.0% (Table 3). Fe is mainly in the form of Fe(OH)_3_, which is the oxidation and hydration product of FeS_2_ in bauxite. Fe(OH)_3_ is unstable under strong alkaline and high-temperature conditions, and easily converts into goethite (FeOOH). In fresh red mud, Fe(OH)_3_ and FeOOH may coexist, and FeO may exist in the form of siderite (FeCO_3_). The Al_2_O_3_ content of red mud ranges from 6.4 to 31.0% (Table 3), and Al_2_O_3_ usually exists in two forms, i.e., NaAlO_2_ and Al(OH)_3_, under strongly alkaline conditions. The CaO content ranges from 2.2% to 48.4% (Table 3), and CaO usually forms aragonite (CaCO_3_) or calcite (CaCO_3_). CaCO_3_ can deposit and crystallize after the introduction of quicklime (CaO) and carbon dioxide (CO_2_) during alumina production. The content of Na_2_O in red mud ranges from 2.3% to 16.2% (Table 3) and exists in pore solutions as free Na^+^. In addition, a series of saline deposits or colloid products, such as Na_2_CO_3_, NaHCO_3_, Na_2_SiO_3_, and NaAlO_2_, are formed during desiccation. The Ti_2_O content ranges from 0.05% to 7.0% (Table 3). Ti_2_O exists as rutile and anatase in bauxite.

### 3.2. The Hydrochemistry of Red Mud Leachate and Its Major Elements

Hydrochemical parameters of the red mud leachate samples include pH, EC, ORP, and ionic strength are summarized in Table 5. The pH of red mud leachate ranges from 12.1 to 12.6, which exceeds the Integrated Wastewater Discharge Standard in China (IWDS) (GB8978-1996). The ionic strength ranges from 66.9 to 484.3 mM, and the EC value of red mud leachate ranges from 4.4 to 51.1 mS/cm which exceeds the standard (EC value less than 2.0 mS/cm) for reverse osmosis water treatment system. The ORP ranges from −110.0 to −29.0 mV, indicating a reducing state of the red mud leachate.

Al, Ca, Na, K, Mg, Si, Cl^−^, F^−^, NO_3_^2−^, and SO_4_^2−^ are the major elements and ions (defined based on the concentrations) in the red mud leachate (Figure 3 and Table 6). The maximum contamination levels (MCLs) of type (III) (for drinking water) groundwater of the Quality Standard for Groundwater in China (China (GW)) (GB/T 14848-2017) and US Environment Protection Agency Groundwater Quality Standard (USEPA (GW)) were used to evaluate the possible harmfulness of leachate to the environment and human health.

The Al concentration in red mud leachate ranges from 118.3 to 1327.4 mg/L, which exceeds the recommended values set by USEPA (GW) (2 mg/L) and China (GW) (0.2 mg/L). Na concentrations in the red mud leachate range from 1200.5 to 10,650.0 mg/L, which is 6002.5 to 53,250 times higher than the recommended value by China (GW) (0.2 mg/L). Additionally, Cl^−^, as one of the main anions found in the leachate solution of red mud, has a concentration ranging from 551.4 to 6588.1 mg/L. This range is 2.2 to 26.4 times higher than the recommended value of USEPA (GW) and China (GW) (250 mg/L). The F^−^ concentration in the leachate ranges from 88.0~299.6 mg/L. which is more than 44 times higher than the recommended value of USEPA (GW) (2 mg/L) and China (GW) (1 mg/L). The concentration of NO_3_^2−^ ranges from 183.2–730.7 mg/L, which is 9.2 to 36.5 times higher than the recommended value of China (GW) (20 mg/L). The concentration of SO_4_^2−^ ranges from 502.5–6593.0 mg/L, which is 2.0 to 26.4 times higher than the recommended value of USEPA (GW) and China (GW) (250 mg/L). Pyrite (FeS_2_) in bauxite is likely the primary source of SO_4_^2−^ [38].

### 3.3. The Minor and Trace Elements in Red Mud Leachate

The minor elements of the red mud leachates mainly include As, B, Ba, Cr, Cu, Fe, Ni, Mn, Mo, Ti, V, and Zn (Figure 4 and Table 7). Elements, such as, Cr, and V, form oxyanions under alkaline conditions (pH >10) and are highly soluble in the water [39]. The concentration ranges of these elements are: 0.2–2.0 mg/L for As, 9.7–163.4 mg/L for B, 0.1–0.5 mg/L for Ba, 0.1–5.9 mg/L for Cr, 0.2–1.6 mg/L for Cu, 0.7–15.0 mg/L for Fe, 0.8–1.0 mg/L for Ni, 0.3–0.6 mg/L for Mn, 1.0–14.5 mg/L for Mo, 0.2–1.8 mg/L for Ti, 0.9–6.3 mg/L for V, and 4.2–37.0 mg/L for Zn. The MCLs of type (III) (for drinking water) groundwater of the Quality Standard for Groundwater in China (China (GW)) (GB/T 14848–2017), US Environment Protection Agency Groundwater Quality Standard (USEPA (GW)), and Integrated Wastewater Discharge Standard (IWDS) (GB8978-1996) were used to evaluate the possible harmfulness of leachate to the environment and human health.

Exceedance was found for most of the minor elements. The concentration of As is 20 to 200 times higher than the MCL of USEPA (GW), China (GW). The concentration of B is 19.4 to 326.8 times higher than the MCL of China (GW) (0.5 mg/L). The Cr concentration is 2 to 118 times higher than the MCL of China (GW) (0.05 mg/L). The Fe concentration is 2.3 to 50 times higher than the MCL of China (GW) and USEPA (GW) (0.3 mg/L). The Ni concentration is 40 to 50 times higher than the MCL of China (GW) (0.02 mg/L). The Mn concentration is 6 to 12 times higher than the MCL of USEPA (GW) (0.05 mg/L). The Mo concentration is 14.3 to 207 times higher than the MCL of China (GW) (0.07 mg/L). The Zn concentration is 4.2 to 37.0 times higher than the MCL of China (GW) (1 mg/L).

Trace elements of the red mud leachate mainly include Ag, Be, Cd, Co, Hg, Li, Pb, Sb, Se, Sr, and Tl (Figure 5 and Table 8). The concentrations of these elements are: 2.0–197.0 μg/L for Ag, 52.0 μg/L for Be (only detected in GX-A2-L), 12.0–172.0 μg/L for Cd, 22.0–88.0 μg/L for Co, 275.0–599.0 μg/L for Hg, 28.0–893.0 μg/L for Li, 170.0–1096.0 μg/L for Pb, 14.0–71.0 μg/L for Sb, 525.0–1359.0 μg/L for Se, 100.0–750.0 μg/L for Sr, and 5.0 μg/L for Tl (only detected in GX-A1-L). Meanwhile, the concentrations of elements Ag, Be, Cd, Co, Hg, Pb, Sb, Se, and Tl are 2.4 to 272.0 times higher than the MCLs of the China (GW) or USEPA (GW). However, the MCLs of IWDS are much higher than those for drinking ground-water. The concentrations of the minor and trace elements, such as As, Cr, Ni, Mn, Zn, Be, Cd, Pb, Hg, and Se exceed the MCLs of IWDS; this indicates that red mud leachate is harmful to the environment and cannot be directly discharged into the river network.

## 4. Discussion

The bauxite source and extraction processes have a strong impact on the chemical and mineralogical composition of the red mud. For example, the variation in Na_2_O contents is partially due to the addition of NaOH. In the Bayer process, NaOH is used to dissolve the Al from Fe-rich aluminite bauxite [40]. However, the sintering and Bayer-sintering combined methods are often adopted to refine alumina from insoluble diaspore or kaolinite type bauxite ores (usually, Al and Si-rich, but low in iron content). Additionally, to enhance the dissolution of alumina and reduce the consumption of alkali, lime is usually added in the high-temperature sintering process. CaCO_3_ can deposit and crystallize by introducing lime (CaO) and carbon dioxide (CO_2_) during alumina production. Therefore, the associated red mud from sintering or Bayer-sintering combined methods is usually Ca-rich but low in iron content.

Additionally, the content of iron oxide (Fe_2_O_3_) in the red mud produced by the Bayer process is higher than that in the red mud produced by the sintering process. In the Bayer process, while extracting alumina from the bauxite using aqueous NaOH, Fe_2_O_3_ is left in the residue. Therefore, the red mud usually presents relatively high contents of Fe_2_O_3_ [41]. Red mud from the sintering process usually contains elevated CaO contents, while red mud produced by the Bayer process contains elevated Na_2_O and Al_2_O_3_ contents (Figure 2). This difference is due to the addition of limestone in lieu of NaOH (or Na_2_CO_3_) in the sintering process. The aluminum recovery from the bauxite ore in the Bayer process is lower than in the sintering alumina process [42]. Therefore, Bayer red mud usually contains more alumina (Al_2_O_3_) than red mud from the sintering process.

A series of saline deposits or colloid products, such as CaCO_3_, Na_2_CO_3_, NaHCO_3_, Na_2_SiO_3_, and NaAlO_2_, are formed during desiccation. However, during the wetting process, these deposits can be easily soluble and cause salinization pollution to the land. The high concentration of alkaline substances (usually pH >12) is the primary reason for the classification of residue as a special industrial material in some countries, such as Australia [43]. The alkaline mineral composition of red mud from refining processes consists of natron-decahydrate (Na_2_CO_3_·10H_2_O), calcite (CaCO_3_), hydrogarnet (Ca_3_Al_2_(SiO_4_)_2_(OH)_4_), sodalite (Na_8_(Al_6_Si_6_O_24_)Cl_2_), cancrinite (Na_6_Ca_2_((CO_3_)_2_Al_6_Si_6_O_24_)·2H_2_O). The following dissolution reactions enable the red mud to become strongly alkaline:Ca_3_Al_2_(SiO_4_)_2_(OH)_4_ + 8H_2_O = 3Ca^2+^ + 2Al(OH)_3_ + 2H_4_SiO_4_ + 6OH^−^(1)
Na_8_(Al_6_Si_6_O_24_)Cl_2_ + 24H_2_O = 8Na^+^ + 6Al(OH)_3_ + 6H_4_SiO_4_ + 2Cl^−^ + 6OH^−^(2)
Na_6_Ca_2_((CO_3_)_2_Al_6_Si_6_O_24_)·2H_2_O + 24H_2_O = 6Na+ + 2Ca^2+^ + 6Al(OH)_3_ + 6H_4_SiO_4_ + 8OH^−^ + 2HCO_3_^−^(3)
Na_2_CO3·10H_2_O (s) + H_2_O = 2Na^+^ + HCO_3−_ + OH^−^ + 10H_2_O(4)
CaCO_3_ (s) = Ca^2+^ + CO_3_^2^^−^(5)

The reactions indicate that NaOH, Na_2_CO_3_, NaHCO_3_, and NaAl(OH)_4_ are the major soluble alkalinity compounds in red mud. These compounds are formed mainly due to the addition of NaOH, CO_2_, and CaO during the alumina production process [4]. The main anions leading to the alkalinity of red mud in the leachate are: OH^−^, CO_3_^2−^, HCO_3_^−^, Al(OH)_4−_, H_2_SiO_4_^2−^, and H_3_SiO_4_^−^, most of which are the dissolution products of sodalite (Na_8_(Al_6_Si_6_O_24_)Cl_2_), cancrinite (Na_6_Ca_2_((CO_3_)_2_Al_6_Si_6_O_24_)·2H_2_O), hydrogarnet (Ca_3_Al_2_(SiO_4_)_2_(OH)_4_), and tricalcium aluminate (Ca_3_Al_2_O_6_). However, the alkaline substances of red mud vary with the refining processes. The Bayer red mud is strongly alkaline [9]. Compared with other processes, more caustic soda is added in the Bayer process, resulting in much higher contents in sodium monoxide (Na_2_O) in the red mud, with approximately 9.7% being detected in the red mud sample of SD-A (Table 2).

In general, the major elements (including Al, Ca, K, Na, Mg, Si, Cl−, F−, NO_3_^2−^, and SO_4_^2−^) all exceeded the recommended value of groundwater quality standards and could increase the risk of salinization of soil and water [7,9]. Al mainly exists as Al(OH)_4−_ under high pH (pH > 12) conditions, and the insoluble Al(OH)_3_ in red mud can easily be converted into soluble Al(OH)_4−_. NaOH is added during the refining process, therefore, a high concentration of Na is expected to be retained or dissolved in the leachate. In some cases, KOH is added during the refining process, such as in sample HN-A-L, and a higher content of K (1016.0 mg/L) in the leachate can be found than in leachates of other red muds (Table 6). Na or K is the top components for salinization of soil and water [9].

As one of the main anions found in the major elements of red mud leachate, Cl^−^ is derived from CaCl_2_. CaCl_2_ is used to precipitate soluble hydroxides, aluminates, and carbonate. By adding CaCl_2_, soluble alkaline substances can be converted to calcite (CaCO_3_), hydrocalumite (Ca_4_Al_2_(OH)_12_·CO_3_), and aluminohydrocalcite (CaAl_2_(CO_3_)_2_(OH)_4_·_3_H_2_O). The insoluble tricalcium aluminate (Ca(AlO_2_)_2_) existing in red mud will participate in forming soluble Al(OH)_4__−_ ions [44,45,46]. This precipitation process aims to lower the pH and alkalinity of the leachate before discharge. The F^−^ is mainly leached from the fluoride in red mud, derived from the bauxite ore. The F^−^ content in red mud leachate is more than 44 times higher than the recommended value of USEPA (GW) and China (GW) (Figure 3, Table 6). The long-term intake of water with high fluoride concentration may cause bone fluorosis, dental fluorosis, osteoporosis, resulting in serious human health issues.

The concentrations of minor and trace elements, such as As, B, Cr, Fe, Ni, Mn, Mo, V, Zn, Ag, Be, Cd, Co, Hg, Pb, Sb, Se, and Tl, exceed the standards. The heavy metal elements, such as Cr, Fe, Ni, Mn, Mo, V, Zn, Ag, Co, Hg, and Pb, derived from the metal and its associated minerals in bauxite ore. These heavy metal elements could accumulate in soil and water. They increase ecological risks to crops, agricultural products, and groundwater, and also endanger human health through the food chain [47]. The impacts of red mud and its associated leachate on the surrounding environment largely depend on the management of red mud. Measures to prevent the leakage or spill of red mud and its leachate should be considered in the design of red mud management facilities, especially in the design of the containment system (especially, the liner) of the red mud reservoirs. Therefore, further studies are needed to investigate the compatibility between red mud leachate and liner materials, as the chemical environment, especially high pH and salinity conditions, could affect the hydraulic behavior and strength of the liner materials [48,49,50].

## 5. Conclusions

This study evaluated the chemical and mineralogical composition of red mud produced in China. Red mud and leachate samples were taken from five management facilities in Shandong, Guangxi, and Henan provinces of China. The chemical and mineralogical compositions of red mud and the major, minor and trace elements in the leachate were evaluated by laboratory analytical methods. The results of geochemical analysis on red mud from the literature were compared for a better understanding of the chemical characteristics of red mud.

Based on the findings of this study, the following conclusions and recommendations are drawn:The major chemical components of red mud are SiO_2_, TiO_2_, Al_2_O_3_, Fe_2_O_3_, CaO, and Na_2_O, and the secondary components are K_2_O, MgO, P_2_O_5_, MnO, Cr_2_O_3_, and SO_3_. The main minerals include quartz, calcite, dolomite, hematite, hibschite, sodalite, anhydrite, cancrinite, and gibbsite.The mineral and chemical compositions of red mud vary over different alumina refining processes and parental bauxite. Diaspore is the primary type of bauxite in China. Among the red mud produced by the Bayer, sintering, and combined methods, levels of Na_2_O and Al_2_O_3_ are higher in the red mud from the Bayer method, and CaO is higher in red muds from the sintering method.The major elements and ions in red mud leachates include Al, Ca, Cl^−^, F^−^, K, Na, Mg, NO_3_^2−^, Si, and SO_4_^2−^. The concentrations of Al, Cl^−^, F^−^, Na, NO_3_^2−^, and SO_4_^2−^ exceed the recommended groundwater quality standard of China by up to 6637 times. These ions are likely to increase the salinization in the soil and groundwater.The minor elements in red mud leachates include As, B, Ba, Cr, Cu, Fe, Ni, Mn, Mo, Ti, V, and Zn, while the trace elements in red mud leachate include Ag, Be, Cd, Co, Hg, Li, Pb, Sb, Se, Sr, and Tl. The concentrations of these elements are 2.4 to 272.0 times higher than the MCLs of China (GW) and USEPA (GW). These heavy metal elements could accumulate in soil and water, increase ecological risks to crops, agricultural products, and groundwater, and endanger human health through the food chain.

## Figures and Tables

**Figure 1 ijerph-16-01297-f001:**
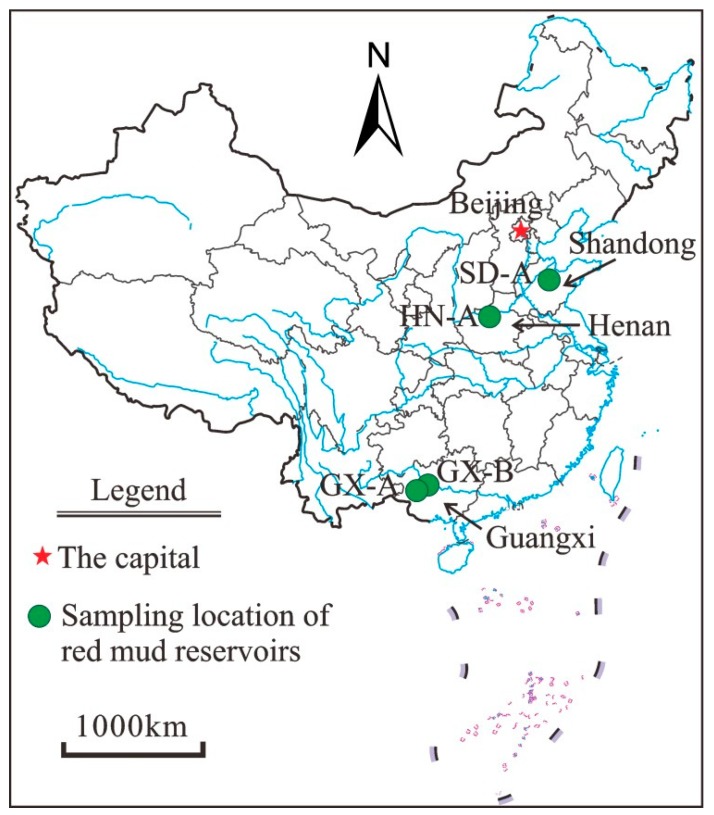
Locations for red mud and leachate sampling of the current study.

**Figure 2 ijerph-16-01297-f002:**
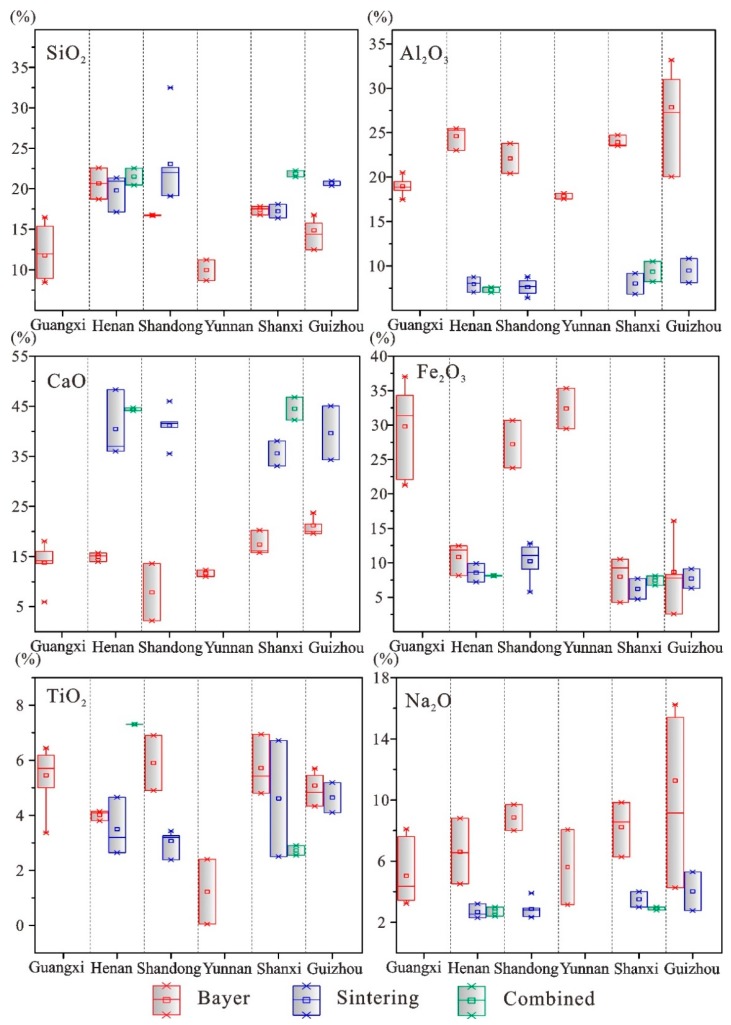
Box plot of chemical composition in Red Mud from the alumina manufactures in China.

**Figure 3 ijerph-16-01297-f003:**
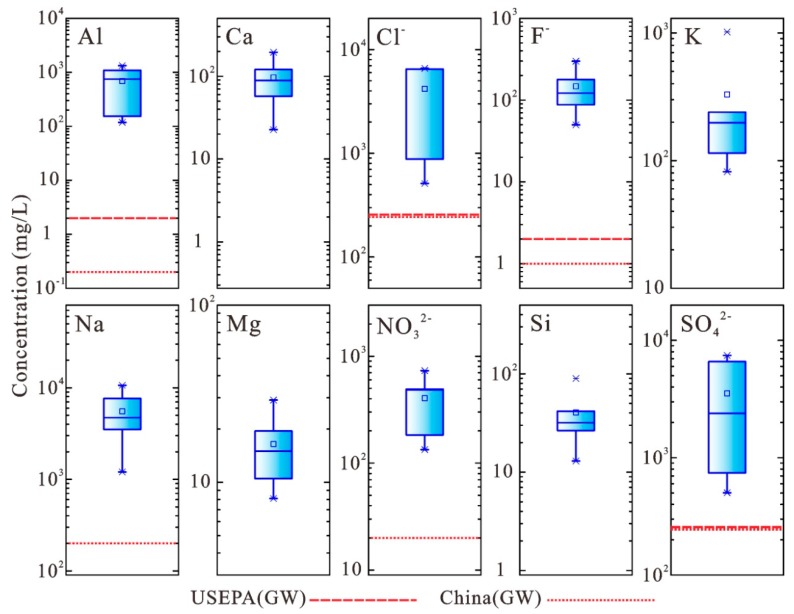
Boxplot of the concentration of major elements/ions in red mud leachate (Red lines indicates the MCLs of groundwater quality standards).

**Figure 4 ijerph-16-01297-f004:**
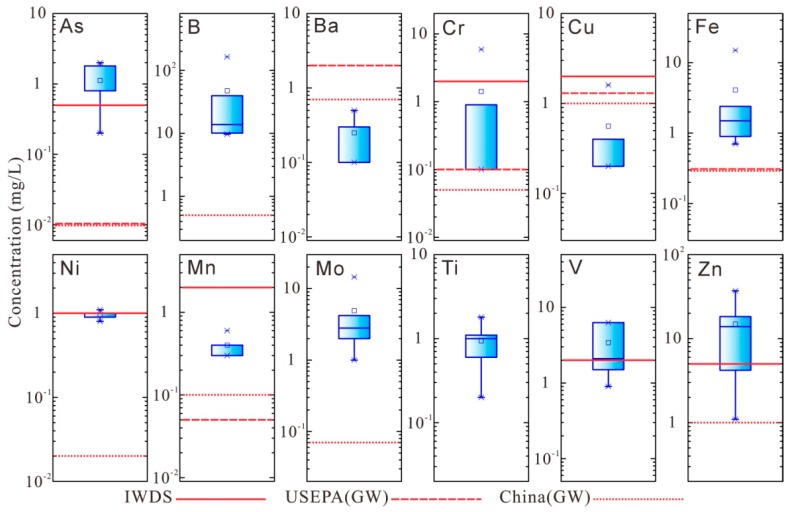
Boxplot of the concentration of minor elements in red mud leachate (Red lines indicates the MCLs of groundwater quality standards).

**Figure 5 ijerph-16-01297-f005:**
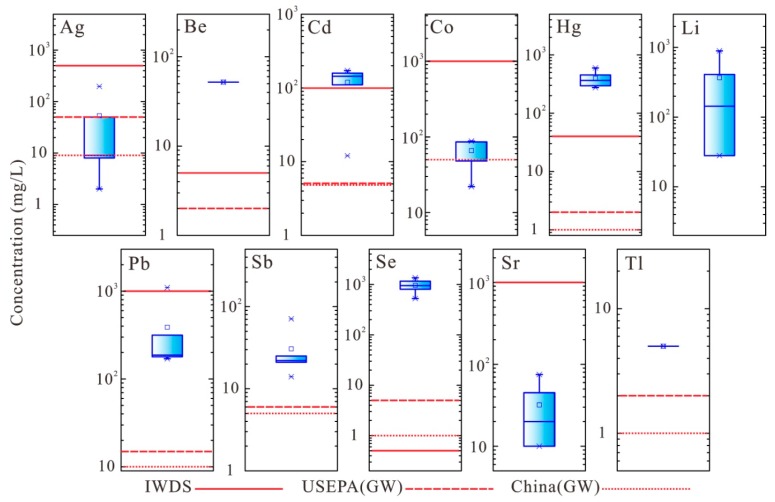
Boxplot of the concentration of trace elements in red mud leachate (Red lines indicates the MCLs of groundwater quality standards).

**Table 1 ijerph-16-01297-t001:** The major chemical compositions (oxide wt.%), loss on ignition (LOI, wt.%) and Al/Si in bauxite over the world.

Bauxite Origin	Al_2_O_3_	Fe_2_O_3_	SiO_2_	TiO_2_	MgO	K_2_O	CaO	Na_2_O	LOI	Al/Si	Ref.
Juruti, Brazil	45.0	14.7	24.9	1.9	-	-	-	-	18.2	1.8	[19]
Mandan, Iran	44.4–64.1	2.8–22.3	3.3–9.6	2.1–3.2	0.1–0.3	0.02–0.3	0.05–11.3	0.02–0.07	11.1–23.2	5.4–15.8	[20]
Kanisheeteh, Iran	20.2–33.4	17.2–34.2	23.6–43.6	3.1–4.8	0.01–0.7	0.02–1.3	0.1–0.4	0.01–0.9	6.0–9.7	0.5–1.4	[21]
Rompin, Malaysia	29.0–44.4	2.5–4.0	28.8–52.5	0.2–0.3	>0.01	0.05–0.1	-	-	13.9–24.1	0.55–1.54	[22]
Kuantan, Malaysia	41.3–43.6	20.7–23.5	3.1–11.9	3.5–4.1	0.08–0.09	-	-	-	22.0–25.2	3.5–14.0	[22]
Johor, Malaysia	43.4–50.6	15.1–17.3	3.7–13.8	2.4–2.3	-	-	-	-	22.8–28.0	3.1–13.7	[22]
Guangxi, China	52.3–62.4	16.6–24.5	3.5–8.3	1.9–3.2	0.02–0.09	0.004–0.03	0.03–0.1	0.001–0.2	11.8–17.7	6.3–17.7	[14]
Henan, China	63.3–69.4	1.5–9.4	8.7–18.0	0.7–3.2	-	-	-	-	12.8–14.7	3.4–7.6	[15]
Shandong, China	37.4	8.7	32.2	2.3	0.9		3.2	0.9	13.7	1.2	[16]
Guangxi, China	58-6	15.0–17.0	5.0–6.0	-	-	-	-	-	-	9.9	[17]
Guizhou, China	67.0–68.0	2.2–3.0	8.8–11.1	-	-	-	-	-	-	6.1–7.8	[17]
Henan, China	64.0–74.0	3.0–5.1	7.5–13.7	-	-	-	-	-	-	4.7–9.4	[17]
Shandong, China	54.0–61.0	5.0–9.0	15.0–22.0	-	-	-	-	-	-	3.7–3.9	[17]
Shanxi, China	63.0–65.0	2.0–3.0	11.0–13.0	-	-	-	-	-	-	5.0–5.6	[17]
Henan, China	66.8	1.4	12.5	3.0	0.1	0.3	0.3	0.04	14.2	5.4	[18]
France	58.6	26.2	0.8	2.8	-	-	-	-	-	73.3	[23]
France	76.4	4.8	0.8	3.3	-	-	-	-	-	95.5	[23]
France	60.6	26	0.3	0.8	-	-	-	-	-	209.0	[23]
France	63.7	5.5	13.3	2.4	-	-	-	-	-	4.8	[23]
Romania	59.7	23.7	1.5	3.1	-	-	-	-	-	2.5	[23]
Romania	65.5	21.3	0.8	2.8	-	-	-	-	-	82	[23]
Italy	57.6	26.6	2.8	1.3	-	-	-	-	-	20.7	[23]
Italy	58.9	18.6	7.9	-	-	-	-	-	-	7.4	[23]
Alabama, USA	58.2	3.6	2.9	3.4	-	-	-	-	-	20.1	[23]
Arkansas, USA	62.3	1.7	2.0	3.5	-	-	-	-	-	31.1	[23]
Arkansas, USA	55.1	6.1	10.1	-	-	-	-	-	-	5.4	[23]
Georgia, USA	64.9	0.3	0.6	1.1	-	-	-	-	-	104.7	[23]
Guyana, UK	64.4	0.5	2.7	0.1	-	-	-	-	-	23.6	[23]
Guyana, UK	70.9	0.8	1	1.1	-	-	-	-	-	70.9	[23]

**Table 2 ijerph-16-01297-t002:** Sampling program of red mud and its associated leachate from the red mud reservoirs in China.

Locations	Red Mud Reservoirs	Types of Samples	Processes Method
Guangxi	3 Reservoirs (denoted as GX-A1, GX-A2, and GX-B)	Bauxite residual, Leachate	Combined and Bayer
Shandong	1 Reservoirs (denoted as SD-A)		Bayer
Henan	1 Reservoir (denoted as HN-A)		Sintering

**Table 3 ijerph-16-01297-t003:** The major chemical compositions (Oxide wt.%) of red mud in China.

Refinery	Process	SiO_2_	Al_2_O_3_	Fe_2_O_3_	CaO	TiO_2_	Na_2_O	MgO	K_2_O	Cr_2_O_3_	P_2_O_5_	SO_3_	MnO	LOI	Refs.
HN-A	Bayer	22.5	25.3	8.1	15.1	3.8	8.8	1.3	1.2	0.1	0.3	-	0.1	12.1	This Study
SD-A	Bayer	16.7	20.4	23.7	13.6	4.9	8.0	0.4	0.2	0.1	0.3	0.4	-	11.2	
GX-A1	Bayer	16.6	23.8	30.6	2.2	6.9	9.7	0.2	0.1	0.1	0.3	0.4	-	9.1	
GX-B	Bayer	15.3	20.5	21.2	14.5	5.7	8.1	0.7	0.1	0.3	0.2	0.7	0.1	12.5	
GX-A2	Bayer	16.4	19.1	22.0	16.0	5.0	7.6	0.9	0.1	0.2	0.2	0.3	0.1	12.0	
Guangxi	Bayer	9.1	18.7	37.0	6.0	6.4	3.4	-	-	-	-	-	-	-	[25]
Guangxi	Bayer	8.4	18.5	31.3	18.1	6.2	3.2	0.7	0.2	0.3	-	0.3	0.7	13.9	[26]
Guangxi	Bayer	11.9	17.5	32.5	14.1	5.5	4.0	-	1.0	-	-	-	-	-	[27]
Guangxi	Bayer	11.9	19.5	29.9	13.9	3.4	4.6	2.8	-	-	-	-	-	11.6	[28]
Guangxi	Bayer	8.9	18.9	34.3	13.6	6.1	4.4	0.4	0.1	-	-	-	-	-	[5]
Henan	Bayer	20.6	25.5	11.8	14.0	4.1	6.6	1.5	2.1	-	-	-	-	-	[5]
Henan	Combined	20.4	7.6	8.2	44.7	7.3	3.0	-	-	-	-	-	-	11.0	[29]
Henan	Sintering	21.4	8.8	8.6	36.0	2.6	3.2	1.9	0.8	-	-	-	-	16.3	[30]
Henan	Bayer	18.6	23.0	12.4	15.7	4.1	4.5	1.6	1.8	-	-	-	-	12.5	[8]
Henan	Combined	22.5	7.0	8.1	44.1	7.3	2.4	2.0	0.5	-	-	-	-	8.3	[30]
Henan	Sintering	17.2	8.1	9.8	37.0	4.7	2.5	1.0	2.5	-	0.3	-	0.0	17.0	[31]
Henan	Sintering	20.9	7.0	7.2	48.4	3.2	2.3	-	-	-	-	-	-	-	[32]
Shandong	Sintering	32.5	8.3	5.7	41.6	-	2.3	-	-	-	-	-	-	-	[5]
Shandong	Sintering	19.1	8.8	12.2	35.5	2.4	3.9	1.9	0.4	-	-	0.3	-	-	[33]
Shandong	Sintering	19.1	6.9	12.8	46.0	3.4	2.4	1.2	1.2	-	-	-	-	5.7	[34]
Shandong	Sintering	22.0	6.4	9.0	41.9	3.2	2.8	1.7	0.3	-	-	-	-	11.7	[30]
Shandong	Sintering	22.7	7.7	11.0	40.8	3.3	2.9	1.8	0.4	-	-	-	-	11.8	[35]
Yunnan	Bayer	11.2	17.5	35.3	12.4	2.4	8.1	1.6	-	-	-	-	-	10.0	[36]
Yunnan	Bayer	8.6	18.2	29.4	11.0	0.1	3.2	-	0.1	-	-	9.3	-	-	[29]
Shanxi	Combined	22.2	10.5	6.8	42.3	2.6	3.0	2.5	0.9	-	-	-	-	-	[5]
Shanxi	Sintering	18.1	9.2	4.7	38.1	6.7	4.0	-	-	-	-	-	-	12.3	[37]
Shanxi	Bayer	17.4	23.6	4.2	20.2	6.9	8.6	-	-	-	-	-	-	11.5	[37]
Shanxi	Bayer	16.7	23.5	9.2	16.1	4.8	6.3	-	-	-	-	-	-	12.2	[37]
Shanxi	Sintering	16.4	6.8	7.6	33.1	2.5	3.0	1.7	0.2	-	-	-	-	-	[24]
Shanxi	Combined	21.4	8.2	8.1	46.8	2.9	2.8	1.7	0.2	-	-	-	-	-	[32]
Shanxi	Bayer	17.8	24.7	10.4	15.7	5.4	9.8	0.9	0.6	-	-	-	-	-	[32]
Guizhou	Bayer	16.7	33.2	8.2	19.6	4.3	16.2	0.7	0.6	-	-	-	-	1.7	[34]
Guizhou8	Bayer	14.3	20.0	16.0	20.0	5.5	9.2	0.9	2.0	-	-	-	-	12.3	[34]
Guizhou	Bayer	15.7	27.3	7.7	21.5	4.8	15.4	1.7	2.8	-	-	-	-	3.8	[34]
Guizhou	Sintering	20.4	10.8	9.1	34.3	4.1	5.3	1.2	5.3	-	-	-	-	14.1	[34]
Guizhou	Bayer	12.4	31.0	2.5	23.7	5.7	4.3	0.7	0.5	-	-	-	-	16.5	[31]
Guizhou	Sintering	21.0	8.1	6.2	45.1	5.2	2.8	-	0.1	-	-	-	-	9.4	[31]

**Table 4 ijerph-16-01297-t004:** The mineral compositions (wt.%) of red mud sampled in this study.

Mineral	Formula	HN-A	SD-A	GX-A	GX-B	SD-B
Quartz	SiO_2_	10	10		-	10
Calcite	CaCO_3_	-	20	10	10	-
Dolomite	CaMg(CO_3_)_2_	-	-	-	-	-
Hematite	Fe_2_O_3_	20	40	35	35	50
Hydrogarnet	Ca_3_Al_2_(SiO_4_)_2_(OH)_4_	40	-	30	30	-
Sodalite	Na_8_(Al_6_Si_6_O_24_)Cl_2_	30	20	-	25	-
Anhydrite	CaSO_4_	-	10	-	-	-
Cancrinite	Na_6_Ca_2_((CO_3_)_2_Al_6_Si_6_O_24_)·2H_2_O	-	-	25	-	-
Gibbsite	Al(OH)_3_	-	-	-	-	40

**Table 5 ijerph-16-01297-t005:** Bulk chemical parameters of red mud leachate samples from the current study.

Leachate Samples	Sample Location	pH	EC@25 °C (mS/cm)	ORP (mV)	Ionic Strength (mM) *
HN-A-L	Leachate drainage	12.6	27.6	−72.0	383.3
SD-A-L		12.6	51.1	−110.0	484.3
GX-A1-L		12.6	4.4	−29.0	66.9
GX-B-L		12.4	26.3	−63.0	238.8
GX-A2-L		12.1	9.71	−46.0	159.5

* Calculated using Visual MINTEQ, charge differences <5%.

**Table 6 ijerph-16-01297-t006:** The concentration of major elements (ions) in the red mud leachate (mg/L).

Samples	Al	Ca	Na	Mg	K	Si	Cl^−^	F^−^	NO_3−_	SO_4_^2−^
HN-A-L	1327.4	22.6	7634.5	8.1	1016.0	31.8	6588.1	299.6	492.2	2386.4
SD-A-L	745.5	57.1	10650.0	10.5	81.8	89.9	6490.5	121.8	483.4	7453.3
GX-A1-L	118.3	120.6	1200.5	19.5	114.4	13.0	511.4	88.0	183.2	502.5
GX-B-L	152.9	194.6	4707.0	29.1	197.6	26.5	6459.8	178.6	730.7	6593.0
GX-A2-L	1095.5	88.9	3506.0	15.0	239.3	41.7	877.1	49.8	133.8	741.7
Range	118.3–1327.4	22.6–194.6	1200.5–10650.0	8.1–29.1	81.8–1016.0	13.8–89.9	511.4–6588.1	88.0–299.6	183.2–730.7	502.5–6593.0
USEPA (GW)	2	-	-	-	-	-	250	2	-	250
China (GW)	0.2	-	200	-	-	-	250	1	20	250

**Table 7 ijerph-16-01297-t007:** The concentration of minor elements (ions) in red mud leachate (unit: mg/L).

Samples	As	B	Ba	Cr	Cu	Fe	Ni	Mn	Mo	Ti	V	Zn
HN-A-L	0.2	163.5	-	0.1	0.2	1.5	0.8	0.4	1.0	1.0	2.1	37.0
SD-A-L	1.8	39.7	0.1	5.9	1.6	15.0	0.9	0.3	14.5	1.8	6.3	18.3
GX-A1-L	2.0	9.7	0.3	0.9	0.4	2.4	1.1	0.6	2.8	0.6	6.3	4.2
GX-B-L	0.8	10.1	0.5	0.1	0.2	0.9	1.0	0.4	4.2	1.1	0.9	1.1
GX-A2-L	0.8	13.8	0.1	0.1	0.4	0.7	1.0	0.3	2.0	0.2	1.5	13.9
Range	0.2–2.0	9.7–163.4	0.1–0.5	0.1–5.9	0.2–1.6	0.7–15.0	0.8–1.0	0.3–0.6	1.0–14.5	0.2–1.8	0.9–6.3	4.2–37.0
USEPA (GW)	0.01	-	2	0.1	1.3	0.3	-	0.05	-	-	-	-
China (GW)	0.01	0.5	0.7	0.05 *	1	0.3	0.02	0.1	0.07	-	-	1
IWDS	0.5	-	-	1.5	2.0	-	1.0	2.0	-	-	2.0	5.0

* MCL of Cr(VI).

**Table 8 ijerph-16-01297-t008:** The concentration of trace elements in red mud leachate (unit: μg/L).

Samples	Ag	Be	Cd	Co	Hg(total)	Li	Pb	Sb	Se	Sr	Tl
HN-A-L	9	/	12	22	599	893	187	14	951	100	/
SD-A-L	50	/	144	48	453	409	1095	71	1160	100	/
GX-A1-L	8	/	172	86	296	/	179	22	801	450	5
GX-B-L	2	/	159	86	275	28	170	21	1359	750	/
GX-A2-L	197	52	110	88	365	143	316	25	525	200	/
Range	2–197	≤52	12–172	22.88	275–599	≤893	170–1095	14–71	525–1359	100–750	≤5
USEPA (GW)	50	2	5	-	2	-	15	6	5	-	2
China (GW)	50	2	5	50	1	-	10	5	10	-	0.1
IWDS	500	5	100	1000	50	-	1000	-	0.5	10,000	-

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
