# Peer review of "Geochemical Characteristics and Toxic Elements in Alumina Refining Wastes and Leachates from Management Facilities"

_ijerph, 2019, doi:10.3390/ijerph16071297_

Round 1

Reviewer 1 Report

This manuscript describe the chemical and mineralogical characteristics of red mud and its leachates. Different procedures of alumina refining were evaluated.

The analytical and bibliographic data presented in the work are interesting and useful for researchers in the field of industrial waste management and reuse.

However, some modifications and improvements must be made.

- Units must be included in Table 1, Table 3 and Table 4.

- Table 2 describes two reservoirs from Guangxi, but 3 names are included (GX-A1, GX-A2 and GX-B).

- A brief description of reservoir can be included: the period during which it has been in operation, size or volume, mean annual rainfall in the area...

- Table 3. The meaning of LOI should be included in the table.

- Table 4. Line of total amount can be deleted. It is a obvious information.

- Page 6 Line 160. Authors say that the addition of NaOH is excessive. Some data or reference must be included to justify this affirmation.

- Fig. 1. Values or quartiles of rectangles and lines should be included in the legend of table

- Table 5. The meaning of RMD should be included in the table. 

- Page 8 Line 208. The sample SD-B does not exist in Table 2 (only SD-A)

- Page 9 Lines 223-224. There is not a positive correlation between pH and Al concentration (pH is similar in all samples (12.1 to 12.6). There is not enough data to support this affirmation.

Page 12 Lines 311-312. "… The heavy metals cannot be easily degraded by microorganisms ...".  They cannot be degraded easily or with difficulties. Heavy metals do not undergo microbiological degradation processes, but chemical processes of precipitation, adsorption ... This phrase must be rewritten

The criteria to evaluate the possible harmfulness (comparison to drinking water and ground water quality  standards of USEPA and China) is not adequate. . Obviously, the leachates present contents of heavy metals much higher than those established for drinking water and therefore their comparison lacks practicality. The red mud leachate is a industrial efluent and other criteria of evaluation should be employed

Author Response

REPLY TO REVIEW

Dear Editor and Reviewers,

Thank you so much for your time and efforts! I, with the other five authors, profoundly appreciate your valuable comments toward the improvement of the paper and your detailed corrections. We have extensively revised our manuscript. Responses to the comments and descriptions of the changes made on the manuscript are given in this file. It should be noted that all page and line numbers in the REPLY TO REVIEW” refer to the revised clean copy of the manuscript.

Thanks and best regards,

Jiannan Chen, PhD 

Reviewer #1:

COMMENT 1: Units must be included in Table 1, Table 3 and Table 4.

RESPONSE:

We apologized for the lacking of the measure unit. The Units (oxide wt.%) have been added into Table 1, Table 3, and Table 4.

COMMENT 2: Table 2 describes two reservoirs from Guangxi, but 3 names are included (GX-A1, GX-A2 and GX-B). A brief description of reservoir can be included: the period during which it has been in operation, size or volume, mean annual rainfall in the area

RESPONSE:

Thank you for your comment. Red mud samples GX-A1 and GX-A2 were collected from the same alumina manufacture but different reservoirs in Pingguo County, and red mud sample GX-B was collected from another reservoir in Jingxi County. The description of sampling location and their weather condition were added into paragraph 2.2.

COMMENT 3: Table 3. The meaning of LOI should be included in the table. -Table 4. Line of total amount can be deleted. It is a obvious information. -Page 6 Line 160. Authors say that the addition of NaOH is excessive. Some data or reference must be included to justify this affirmation. -Fig. 1. Values or quartiles of rectangles and lines should be included in the legend of table- Table 5. The meaning of RMD should be included in the table. - Page 8 Line 208. The sample SD-B does not exist in Table 2 (only SD-A)

RESPONSE:

Thank you for pointing out the confusions. We have extensively revised the manuscript. The meaning of LOI (“LOI: loss on ignition”) have been added in the caption of Table 1 (first time appears in the text). In Table 4, the line of total amount has been deleted. A reference has been added to justify the sentence ”The abundance of Na2O in red mud is caused by the excessive addition of NaOH in alumina production.” 

The reference is number 40:

40. Borra, C.R.; Blanpain, B.; Pontikes, Y.; Binnemans, K.; Van Gerven, T. Recovery of Rare Earths and Major Metals from Bauxite Residue (Red Mud) by Alkali Roasting, Smelting, and Leaching. J. Sustain. Metall. 2017, 3, 393–404. https://doi.org/10.1007/s40831-016-0103-3

The RMD in Table 5 has been deleted because this data is not relevant to the scope of the current manuscript. The sample ”SD-B” has been modified to “SD-A”. Fig. 1 (now presenting as Fig. 2) has been modified as below:

Figure 2. Box plot of the chemical composition in Red Mud from the alumina manufactures in China

COMMENT 4: Page 9 Lines 223-224. There is not a positive correlation between pH and Al concentration (pH is similar in all samples (12.1 to 12.6). There is not enough data to support this affirmation.

RESPONSE:

Thank you for pointing out the inappropriate statement. This sentence “The Al concentration showed a positive correlation with the pH of red mud leachate under higher pH conditions.” has been removed by the authors.

COMMENT 5: Page 12 Lines 311-312. "… The heavy metals cannot be easily degraded by microorganisms ...".  They cannot be degraded easily or with difficulties. Heavy metals do not undergo microbiological degradation processes, but chemical processes of precipitation, adsorption ... This phrase must be rewritten

RESPONSE:

Thank you very much for your comments. “These heavy metals cannot be easily degraded by microorganisms and have potential effects on human health through biological chains” has been rewritten in 308-310 as “They increase ecological risks to crops, agricultural products, and groundwater, and also endanger human health through the food chain [48].”

COMMENT 6: The criteria to evaluate the possible harmfulness (comparison to drinking water and ground water quality  standards of USEPA and China) is not adequate. . Obviously, the leachates present contents of heavy metals much higher than those established for drinking water and therefore their comparison lacks practicality. The red mud leachate is a industrial efluent and other criteria of evaluation should be employed

RESPONSE:

Thank you very much for providing the professional comments. The Integrated Wastewater Discharge Standard in China (IWDS) (GB8978-1996)  has been employed to evaluate the minor and trace elements in red mud leachate. Please find the results in Table 7 & 8, and Fig. 5 & 6.

Reviewer 2 Report

The tpoic of this work is intersting, the presentation is good, but in general the quality of this study needs improvement. Some major comments are given below: 

1) Abstarct needs to be more condensed.

2) The Refs in Tables muste ne palces in separate column in the same row of the sample.

3) The boxplots must be bigger (they are not easily readable).

4) Cocnlusions need to be more condensed.

5) The discussion should be re-written. It seems to be just a report of the experimental findings without deep analysis.

6) English must be improved. Too many typos and grammar (syntax) errors

7) The novelty musy be further emphasized.

Author Response

REPLY TO REVIEW

Dear Editor and Reviewers,

Thank you so much for your time and efforts! I, with the other five authors, profoundly appreciate your valuable comments toward the improvement of the paper and your detailed corrections. We have extensively revised our manuscript. Responses to the comments and descriptions of the changes made on the manuscript are given in this file. It should be noted that all page and line numbers in the REPLY TO REVIEW” refer to the revised clean copy of the manuscript.

Thanks and best regards,

Jiannan Chen, PhD

Reviewer #2:

COMMENT 1: Abstract needs to be more condensed.

RESPONSE:

Thank you for your comment. The abstract has been condensed as “A nationwide investigation was carried out to evaluate the geochemical characteristics and environmental impacts of red mud and leachates from the major alumina plants in China. The chemical and mineralogical compositions of red mud were investigated, and the major, minor, and trace elements in the leachates were analyzed. The mineral and chemical compositions of red mud vary over refining processes (i.e., Bayer, sintering, and combined methods) and parental bauxites.  The main minerals in the red mud are quartz, calcite, dolomite, hematite, hibschite, sodalite, anhydrite, cancrinite, and gibbsite. The major chemical compositions of red mud are Al, Fe, Si, Ca, Ti, and hydroxides. The associated red mud leachate is hyperalkaline (pH>12), which can be toxic to aquatic life. The concentrations of Al, Cl-, F-, Na, NO32-, and SO42- in the leachate exceed the recommended groundwater quality standard of China by up to 6637 times. These ions are likely to increase the salinization of the soil and groundwater. The minor elements in red mud leachate include As, B, Ba, Cr, Cu, Fe, Ni, Mn, Mo, Ti, V, and Zn, and the trace elements in red mud leachate include Ag, Be, Cd, Co, Hg, Li, Pb, Sb, Se, Sr, and Tl. Some of these elements have the concentration up to 272 times higher than those of the groundwater quality standard and are toxic to the environment and human health. Therefore, scientific guidance is needed for red mud management, especially for the design of the containment system of the facilities.

COMMENT 2: The Refs in Tables must be places in separate column in the same row of the sample.

RESPONSE:

We apologized for the format error. The references in Table 1 and Table 3 have been placed in a separate column.

COMMENT 3: The boxplots must be bigger (they are not easily readable).

RESPONSE:

Thank you for your suggestion. The figures have been modified, and the resolution has been improved resolution to make them more readable.

COMMENT 4: The discussion should be re-written. It seems to be just a report of the experimental findings without deep analysis.

RESPONSE:

Thank you for your comment. The original ”Results and Discussion” section has been split into a “Results” section and a “Discussion” section. The “Discussion” section has provided a more in-depth analysis on the results.

COMMENT5: English must be improved. Too many typos and grammar (syntax) errors

RESPONSE:

We apologized for the typos and grammar (syntax) errors. The authors have checked the article and modified the errors.

Reviewer 3 Report

Overall the manuscript is well written and interesting. This is a timely piece of work which will be of great interest internationally, given the significance of red mud waste the environmentally appropriate storage of the waste is key to preventing toxicity etc., however, I have some reservations about the manuscript before it is publishable.

General comments:

The first issue is the combination of the result and discussion. This is not the appropriate way to present data. It makes it very difficult to read and process. In my opinion, before the manuscript is accepted this would have to be split into a results section then a separate discussion section, which will provide much greater clarity. The text is already there is just needs to be unraveled.

I feel the authors could go further with their discussion to include comments on any research that has already been done on liners for red mud storage sites. It would also be important to include any policies that have been written to address that storage of red mud after the introduction of the ‘convention on the prevention of marine pollution by dumping of wastes and other matter’.

A figure which outlines the localities in China would be useful for the international reader.

Table 1 is very difficult to read and up to 4 significant figures, these should all be brought to 2 significant figures, depending on your analytical precision.  

Note the use of capital A for alumina is incorrect. The authors also need to address the use of a capital for Bayer (correct) but sintering and combined should have a lower case letters.

Do not use ~ for a range, this should have an ‘m hyphen’ e.g. 1–5

Lines 186-187 use aluminium and alumina interchangeably, please check the manuscript carefully to avoid this.

Could the figures be provided in higher resolution, this would make them easier to read.

I suggest a subheading in the discussion for comparison of guidelines with values found in the ground water.

The conclusions are in the wrong font size.

The reference list need significant revision, as there are multiple duplicates and need to list all authors. There are quite a few reference to what may be theses? This needs to be clarified.

Specific comments these are included on the pdf.

Best of luck, it’s a good paper and with some further work will make a great contribution to the literature.

Eimear Deady

British Geological Survey.

Author Response

REPLY TO REVIEW

Dear Editor and Reviewers,

Thank you so much for your time and efforts! I, with the other five authors, profoundly appreciate your valuable comments toward the improvement of the paper and your detailed corrections. We have extensively revised our manuscript. Responses to the comments and descriptions of the changes made on the manuscript are given in this file. It should be noted that all page and line numbers in the REPLY TO REVIEW” refer to the revised clean copy of the manuscript.

Thanks and best regards,

Jiannan Chen, PhD

Reviewer #3:

COMMENT 1: The first issue is the combination of the result and discussion. This is not the appropriate way to present data. It makes it very difficult to read and process. In my opinion, before the manuscript is accepted this would have to be split into a results section then a separate discussion section, which will provide much greater clarity. The text is already there is just needs to be unraveled.

RESPONSE:

We apologized for the unclear structure of the “Results and Discussion” section in this article. It has been unraveled and split into a “Results” section and a “Discussion” section. In the Discussion section, the characteristic of chemical and mineralogical composition of red mud is analyzed based on the bauxite source and extraction process. Additionally, the potential threatens by the toxic elements, such as the alkaline substances, major, minor and trace elements in red mud leachate, to the environment and human health are analyzed.

COMMENT 2: A figure which outlines the localities in China would be useful for the international reader.

RESPONSE:

Thank you for your comment. The sketch map of sampling location of red mud reservoirs has been made to clearly present the sampling locations (GX-A,GX-B,SD-A, and HN-A).

Figure 1. Locations for red mud and leachate sampling of the current study.

COMMENT 3: Note the use of capital A for alumina is incorrect. The authors also need to address the use of a capital for Bayer (correct) but sintering and combined should have a lower case letters.

Do not use ~ for a range, this should have an ‘m hyphen’ e.g. 1–5

Lines 186-187 use aluminum and alumina interchangeably, please check the manuscript carefully to avoid this.

Could the figures be provided in higher resolution, this would make them easier to read.

The conclusions are in the wrong font size.

RESPONSE:

We apologized for the typos and grammar (syntax) errors. The authors have checked and modified these errors. And the “~” used for a range in this article have been replaced as  ”-”. The “aluminum” are replaced as “alumina”. Figures have been replotted in a higher resolution.

The font size of the conclusions section have been modified.

COMMENT 4: The reference list need significant revision, as there are multiple duplicates and need to list all authors. There are quite a few reference to what may be theses? This needs to be clarified.

RESPONSE:

We apologized for these errors in the reference list. The reduplicated references have been deleted and following references have been added into the list.

Liu, Y.; Ni, W.; Huang, X.; Li, D.; Ma, X. Hydraulic cementitious properties of bayer process red mud from Guangxi Pingguo aluminum corporation. Metal Mine. 2016, 7, 193-196. (in Chinese)

1.       Liao, C.; Lu, H.; Qiu, D.; Xu, X. Recovering valuable metals from red mud generation during alumina production. Light Metal. 2003, 10, 18-22. (in Chinese)

2.       Liu, S.; Xie, G.; Li, R.; Yu, Z. Comprehensive utilization of the red mud from alumina plant. Mining & Metallurgy. 2015, 24(3), 72-75. (in Chinese)

3.       Wang, K.; Li, A.; Deng, H.; Zhu, G. Physicochemical properties of red mud in Shanxi. Light Metal. 2012, 4, 25-28. (in Chinese)

4.       Sun, X.; Ning, P.; Tang, X. L.; Yi, H.; Zhou, L.; Li, K. Heavy metals pollution assessment in soil surrounding Red Mud Ponds in Shaanxian, Henan. Journal of Northwest A&F University (Nat. Sci. Ed.). 2015, 43(5), 122-128. (in Chinese with English abstract)

COMMENT 5: I suggest a subheading in the discussion for comparison of guidelines with values found in the ground water.

RESPONSE:

Thank you very much for your suggestion. The comparison of guidelines with values in the ground water standards and Industry wastewater standards have been placed in the discussion. The comparison shows some of the major, minor, and trace elements in red mud leachate could threaten the environment and human health.

COMMENT 6: Specific comments these are included on the pdf.

RESPONSE:

Thank you so much for your elaborate efforts to point out the errors in this article. We have extensively revised the manuscript. Based on your comments in the pdf, the authors has carefully addressed the comments, and fixed the errors.

Round 2

Reviewer 1 Report

The manuscript was improved  substantially  and I think i can be publkished in present form

Reviewer 2 Report

Accept as it is

Reviewer 3 Report

Well done on your quick revisions, I am glad that the review was useful to you. It is now much clearer to read the results and discussion and the figures are clearer.